# Job Insecurity and Safety Behaviour: The Mediating Role of Insomnia and Work Engagement

**DOI:** 10.3390/ijerph18020581

**Published:** 2021-01-12

**Authors:** Xinyong Zhang, Chaoyue Zhao, Zhaoxiang Niu, Shike Xu, Dawei Wang

**Affiliations:** 1Department of Applied Psychology, Guangdong University of Foreign Studies, Guangzhou 510420, China; zhangxinyong@gdufs.edu.cn; 2School of Psychology and Cognitive Science, East China Normal University, Shanghai 200062, China; 52203200019@stu.ecnu.edu.cn; 3School of Psychology, Shandong Normal University, Jinan 250014, China; 2019305027@stu.sdnu.edu.cn; 4Department of statistics, University of Connecticut, Mansfield, CT 06269, USA

**Keywords:** job insecurity, safety behaviour, insomnia, work engagement

## Abstract

From the perspective of resource conservation theory, this study selected 568 enterprise employees as subjects and conducted data collection using a random sampling method to explore the relationship between job insecurity and safe behaviours as well as the role of insomnia and job engagement in this relationship. The results show that (1) job insecurity is negatively correlated with safety behaviour, (2) insomnia mediates the relationship between job insecurity and safety behaviour, (3) work engagement plays a mediating role in the relationship between job insecurity and safety behaviour, and (4) insomnia and work engagement play a serial mediating role in the relationship between job insecurity and safety behaviour.

## 1. Introduction

With the development of the economy and technology over the last few decades, the safety condition of employees in the workplace has greatly improved [1]. However, job insecurity is still an issue and poses a critical problem in several high-risk industries such as petrochemical enterprises. A considerable number of issues have occurred in petrochemical industries, leading to enormous losses [2]. Safety accidents can cause unpredictable damage to organizations and individuals, which may be caused by improper behaviours of employees [3]. Therefore, it is necessary to strengthen the exploration of employees’ safety behaviour and its influencing factors to effectively formulate intervention measures to improve personal safety behaviour, reduce the incidence of enterprise safety accidents and increase workplace safety.

Safety behaviour refers to employees’ behaviour in complying with the corresponding regulations when faced with risky situations [4]. Studies have shown that organizational and environmental factors, such as high work environment pressure, high workload, high uncertainty, high task complexity, and high requirement emphasizing process and speed of standard operation, can affect employees’ safety behaviours, while the influence of individual factors on employees’ safety behaviours needs to be further explored [5]. Studies have indicated that employees’ inner factors, such as emotional and subjective feelings, play a key role in the formation of unsafe behaviours [6,7]. The feeling of job insecurity, as a typical negative subjective emotion in the workplace, has aroused concern about its potential influence on unsafe behaviours [8]. The feeling of job insecurity is defined as the sense that the consistency and stability of one’s current job is threatened [9]. Does this subjective perception affect employees’ safe behaviour in the workplace? Many studies have argued that unsafe behaviour is ascribed to job insecurity [8,10,11]; however, few studies have examined their mechanisms. Our study further explores the relationship and mechanism between job insecurity and unsafe behaviour.

Employees’ uncertainty about job safety includes concerns about job consistency, which places employees in a stressful situation [12]. Consequently, employees’ sleep quality is affected, and insomnia and sleep disorder problems are present [13]. Self-regulatory resource theory argues that sleeplessness hinders the recovery that can be obtained through self-regulatory resources [14]. With the absence of abundant resources, employees are less likely to engage in safety management and regularize their behaviour to fit safety criteria. Therefore, we suggest that sleeplessness serves as a mediation effect between job insecurity and unsafe behaviour.

Since job insecurity means that employees feel threatened by uncertainty, when job insecurity is present, employees may be driven to despair and burnout [9]. The continuous consumption of self-regulatory resources results from such negative emotional feelings and leads to lower work engagement [15]. Meanwhile, employees’ self-regulatory resources fail to recover from their shortage of sleep, which further affects work engagement in a negative way [16]. Work engagement, which serves as a positive feeling in the work context, is proven to be closely related to employees’ safety behaviour [17]. Low work engagement caused by job insecurity and insomnia indicates that employees are low energy and lack concentration in their work [18]. Such low energy and low concentration deprive employees of their abilities to control behaviour and to participate in safety management. Therefore, we speculate that job engagement also plays a mediating role between job insecurity and safety behaviours.

## 2. Theory and Hypotheses

### 2.1. Job Insecurity and Safety Behaviours

The accelerating development of technology and the globalization of markets have brought about many advantages for society, and they present more rigorous requirements for employees. It is common for a large proportion of people to feel a lack of safety in their job [19]. Job insecurity refers to employees’ expectation of whether there is a risk for the continuity of work and their worries and insecurities about the existing work, which are based on their perception and interpretation of the work environment [9]. Organizational models of job insecurity (such as Organizational model, Social exchange model) and stress model (such as ISR model, Johoda’s latent functions approach to employment, Freyer’s agency restriction approach) have fully predicted and analyzed the influence of job insecurity on organizational and individual physical and mental health [19,20]. Based on the stress model of job insecurity, as a powerful source of stress, job insecurity can lead to a variety of negative outcomes, such as decreased job satisfaction, job performance, job engagement, subjective well-being, increased psychological stress, turnover intention, depression and anxiety [21], and it can significantly affect mental and physical health in the long run [22]. Studies have found that job insecurity impacts the happiness obtained from work by threatening income, social status and social relationships [23,24]. According to the conservation of resources theory, people are always making active efforts to maintain, protect and construct what they consider to be valuable resources, and the potential or actual loss of such resources is a threat to them [25]. Therefore, employees tend to spend more resources confronting threatened feelings [23]. At the same time, the loss of this type of resource also causes a certain loss of employees’ psychological resources [26,27]. In the context of lack of psychological resources, employees will have difficulty coping with the complex and changing environment and dangerous events in the workplace and reduce the compliance with safety regulations [28]. They also tend to reduce behaviours that are beneficial to the organization [29], such as safety behaviours.

Currently, few studies have explored the relationship between job insecurity and safety behaviour. Probst and Brubaker’s (2001) study of food processing workers showed that employees’ lack of sense of security at work can reduce safety behaviour, which thus increases the accident rate [8]. Emberland and Rundmo’s (2010) research on adult citizens of Norway indicates that employees’ job insecurity has a negative impact on safety behaviour through a reduction in employees’ subjective well-being [11]. With the accelerating increase in employees’ job insecurity, society urgently requires a deep investigation into the relationship among the above factors. In high-risk industries, especially petroleum industries, security incidents often cause enormous losses [2]. Therefore, the purpose of this study is to explore the relationship between employees’ sense of security and safe behaviour based on conservation of resources theory.

Neal, Griffin, and Hart (2000) concluded that safety behaviours include two dimensions: safety compliance and safety participation [30]. Safety compliance means that employees can carry out production operations according to the enterprise’s safety policy. Safety participation means that they can spontaneously engage in behaviours that are conducive to corporate safety at work, including behaviours such as assisting colleagues, attending safety conferences, and taking the initiative to improve safety in the workplace. Many scholars have validated this classification and confirmed the essential difference between safety compliance and safety participation [31]. Based on the conservation of resources theory, we believe that safety participation requires employees to be more proactive regarding safety compliance [28]. Meanwhile, previous studies seldom examine the two dimensions of safety behaviour or investigate safety compliance and safety participation separately. Consequently, we cannot draw a clear conclusion from the previous studies. In summary, we believe that job insecurity influences the two dimensions in different ways and propose the following hypotheses:

**Hypothesis** **1.**
*There is a negative correlation between job insecurity and safety compliance.*


**Hypothesis** **2.**
*There is a negative correlation between job insecurity and safety participation.*


### 2.2. The Mediating Effect of Insomnia

Sleep is a basic human need. Society has witnessed increasing insomnia problems. Less sleep time and poor sleep quality have become common problems for many. At the same time, the research on factors related to sleep has also attracted great attention. Regarding the causes of insomnia, research has shown that it is not only affected by objective factors such as gender, age, sleeping environment, and stress events, but it is also closely related to an individual’s physical and psychological factors [32]. In the workplace, studies have shown that employees’ work pressure is significantly positively correlated with insomnia and sleep deprivation [33]. Individuals in stressful work situations often suffer from health problems such as insomnia [13]. Most scholars identify job insecurity as an obstructive stressor [34]. By definition, job insecurity refers to an employee’s fear and worry about becoming unemployed. This fear and worry will cause employees to experience tremendous psychological stress after work [35], which can lead to insomnia at night.

The relationship between insomnia and safety behaviour can be explained by self-regulatory resource theory. The theory posits that individuals use self-regulatory resources to regulate their behaviour [36]. Safety behaviour refers to the specific behaviours adopted by employees to protect themselves, colleagues, and organizations from occupational accidents. Such behaviour includes self-regulation. On the one hand, companies generally formulate safety-related rules and regulations, and employees must adjust their behaviour to comply with them [30]. On the other hand, when people adopt certain difficult and uncomfortable behaviours, they also must consume self-regulation resources [36]. For example, in some high risk workplaces, employees often must perform tedious and uncomfortable safety tasks (e.g., conducting safety inspections, wearing safety equipment), which, to a certain extent, affects the work process of employees or hinders their smooth activities, and consumes their own self-regulation resources [37]. In addition, workers must frequently conduct self-monitoring to check whether their behaviour complies with safety regulations or whether they can cope with the complex and changing external environment [38]. This evidence indicates that employees require many self-regulatory resources to perform and maintain their own safe behaviours.

Sleep, as a necessary homeostatic process for individuals to restore energy and cognitive resources, plays a vital role in the restoration of self-regulating resources [14]. Through adequate sleep, employees can obtain and supplement a large amount of self-regulatory resources to cope with a complex working environment and constantly monitor whether their behaviour meets safety regulations [37]. Previous studies have shown that employees with sleep problems exhibit low performance, high absenteeism, and high accident rates [39,40]. Especially in some high-risk industries, employees’ sleep problems can significantly lead to the incidence of occupational safety accidents [41]. At the same time, sleep problems can cause individuals to feel drowsiness at work [42]. Drowsiness can cause employees to become paralyzed at work, thereby reducing their own safety behaviours. Based on the theory of self-regulatory resources, this study concluded that insomnia hinders the recovery of employees’ self-regulation resources and reduces the performance of employees’ self-regulation behaviours, thereby reducing employees’ compliance with safety regulations [43,44]. It is difficult for employees to actively participate in safe work practices. Therefore, we propose the following hypotheses:

**Hypothesis** **3.**
*Insomnia plays a mediating role in job insecurity and safety compliance.*


**Hypothesis** **4.**
*Insomnia plays a mediating role in job insecurity and safety participation.*


### 2.3. The Mediating Effect of Job Engagement

As a powerful source of stress, job insecurity will reduce the resources that employees put into their work and result in a lack of energy and enthusiasm for their work [45]. The relationship between job insecurity and job engagement has been verified by most scholars. Greenhalgh (1984) found that employees with work insecurity cannot devote themselves to their work [46]. De Witte (2005) shows that when employees feel insecure at work, their work enthusiasm will decrease, and their work engagement will be reduced [47]. The research by Guarnaccia et al., (2018) shows that employees’ job insecurity will reduce their sense of self-efficacy, which will negatively affect their work engagement [15].

Work engagement generally refers to individual employees devoting themselves to work based on their psychological interest and curiosity in their work and being willing to sacrifice their rest time and offer extra labour. It reflects the degree of personal sense of identity or the importance of an individual’s self-image evaluation regarding work [18]. If employees have low work engagement, it means that they may suffer from a lack of focus and enthusiasm for their work [18]. Especially in certain dangerous workplaces, distraction will cause employees to become unable to devote themselves to their work and to ignore safety issues, which makes it difficult to concentrate on implementing safety procedures [48]. At the same time, employees with low work engagement lack enthusiasm and energy to cope with the complex and changing working environment, and they will not actively provide the necessary security support and assistance for colleagues or organizations [48]. Based on the above inferences, we propose the following hypotheses:

**Hypothesis** **5.**
*Work engagement plays a mediating role in job insecurity and safety compliance.*


**Hypothesis** **6.**
*Work engagement plays a mediating role in job insecurity and safety participation.*


### 2.4. The Serial Mediating Role of Insomnia and Job Engagement

In summary, we speculate that insomnia and work engagement have a certain mediating effect between job insecurity, safety compliance and safety participation; however, whether they are parallel or serial mediating relationships remains to be further verified.

Employees’ job insecurity represents their fears and worries about the consistency of their work [9]. Such worries will infiltrate their lives outside of work, affect their sleep quality at night, and lead to insomnia. Sleep, as a basic requirement for human functioning, plays an important role in energy preservation and nervous system recovery [49]. Poor sleep quality will reduce the chances of fatigue recovery and increase drowsiness [50]. This will cause employees to be in a state of exhaustion of resources in the morning, making it difficult for them to concentrate on their work [50], in turn affecting their degree of work engagement during the day [51,52]. Furthermore, a low level of work engagement means low concentration and a low energy level. Low concentration will cause employees to ignore safety regulations and reduce compliance with safety rules and regulations; low energy and enthusiasm will prevent employees from actively participating in workplace safety [17]. Therefore, we have reason to state that job insecurity first leads to insomnia in employees and then reduces work engagement, which ultimately leads to a decrease in safety behaviour. That is, insomnia and work engagement play a serial mediating role between job insecurity and safe behaviour. Based on the above inferences, we propose the following hypotheses:

**Hypothesis** **7.**
*Insomnia and work engagement play a serial mediating role in job insecurity and safety compliance.*


**Hypothesis** **8.**
*Insomnia and work engagement play a serial mediating role in job insecurity and safety participation.*


### 2.5. Research Model

This study proposes the following models (see Figure 1) based on the theory and analysis of previous studies:

## 3. Method

### 3.1. Sample and Procedures

The participants in this study were 568 employees from an oil enterprise located in China. All procedures performed in this study involving human participants were conducted in accordance with the ethical standards of the Academic Board of Shandong Normal University and with the 1964 Helsinki Declaration and its later amendments or comparable ethical standards. The ethical approval project identification code is SDNU2020054. Informed consent was obtained from leaders and employees of each company. The information of all the participants was kept strictly confidential, with each participant reserving the right to withdraw from the study at any time.

In this study, SEM statistical techniques were used to determine the relationship between the research hypotheses. A total of 605 questionnaires were distributed, and 568 were returned (a 93.9% response rate). Among these participants, 43.5% (*n* = 247) were female, 56.5% (*n* = 321) were male, 91.7% (*n* = 521) were married, 4.9% (*n* = 28) were unmarried, and 3.3% (*n* = 19) were other. In terms of age, 1.1% (*n* = 6) were under 25 years old, 7.9% (*n* = 45) were 26–30 years old, 19.4% (*n* = 110) were 31–35 years old, 15.3% (*n* = 87) were 36–40 years old, 51.4% (*n* = 292) were 41–50 years old, and 4.9% (*n* = 28) were over 50 years old. In terms of academic qualifications, 1.2% (*n* = 7) had a junior high school diploma, 41.5% (*n* = 236) had a high school diploma, 25.5% (*n* = 145) had an associate degree, 28.9% (*n* = 164) had an undergraduate diploma, and 2.8% (*n* = 16) had a masters’ diploma. In terms of working years, 0.5% (*n* = 3) of the participants had worked for less than 1 year, 2.3% (*n* = 13) had worked for 1–3 years, 1.4% (*n* = 8) had worked for 4–6 years, 12.7% (*n* = 72) had worked for 7–9 years, and 83.1% (*n* = 472) had worked for more than 10 years.

### 3.2. Measures

#### 3.2.1. Job Insecurity Scale 

Job insecurity (see Table A1) was measured with the seven-item scale developed by Staufenbie and König (2011) and translated by Hu (2008) [20,53]. The scale contains two dimensions: cognitive job insecurity and affective job insecurity. Cognitive job insecurity includes four items, such as “My job is secure”, and affective job insecurity includes three items such as “The thought of losing my job troubles me”. All items were rated on a seven-point scale ranging from 1 (Strongly Disagree) to 7 (Strongly Agree). The Cronbach’s alpha coefficient for this scale in our sample was 0.86.

#### 3.2.2. Insomnia Scale

Insomnia (see Table A2) was measured using a four-item scale from the Insomnia Survey (Jenkins, Stanton, Niemcryk, & Rose, 1988) [54]. The items asked the respondents “In the past month, how many days did you have trouble falling asleep,” “… have trouble staying asleep (including waking up too early),” “… wake up several times during the night,” and “… wake up after your usual amount of sleep feeling tired and worn out?” Each item was assessed using a five-point scale that ranged from 1 (not at all) to 5 (more than 14 days). The Cronbach’s alpha coefficient for this scale in our sample was 0.93.

#### 3.2.3. Work Engagement Scale

The work engagement scale (see Table A3) was developed by Schaufeli et al., (2002) and translated by Wang et al., (2015), which contains 17 items [18,55]. The scale consists of three dimensions: vigour, dedication and absorption. Vigour includes six items such as “When I get up in the morning, I feel like going to work”. Dedication includes five items such as “My job inspires me”. Absorption includes six items such as “When I am working, I forget everything else around me”. The scale was measured using a five-point Likert format ranging from 1 (Strongly Disagree) to 5 (Strongly Agree). The Cronbach’s alpha coefficient in our study for this scale was 0.96.

#### 3.2.4. Safety Behaviour Scale

We assessed safety behavior (see Table A4) with an eleven-item measure developed by Neal and Griffin (2006) and translated by Ye et al., (2014) [56,57]. The scale contains two dimensions: safety compliance and safety participation. Safety compliance includes six items such as “I use all the necessary safety equipment to do my job”. Safety participation includes five items such as “I promote the safety program within the organization”. All items were rated on a seven-point scale ranging from 1 (Strongly Disagree) to 7 (Strongly Agree). The Cronbach’s alpha coefficient in our study for this scale was 0.94.

## 4. Results

### 4.1. Common Method Bias

Harman’s single factor test was carried out by exploratory factor analysis. The results showed that there were seven eutectoid factors, and the interpretation rate of the population variance was 77.86%. The interpretation rate of the first common factor was 38.32%. Therefore, there is no serious common method bias in this study [58,59] (Podsakoff, Mackenzie, Lee, & Podsakoff, 2003; Zhou & Long, 2004).

### 4.2. Confirmatory Factor Analyses

We used Mplus 7.0 (Beijing Tianyan Rongzhi Software Co., Ltd., Beijing, China) to conduct confirmatory factor analyses (CFA). The hypothesized five-factor model (*χ*^2^ (659) = 1893.84, *p* < 0.001, root-mean-square error of approximation (RMSEA) = 0.06, comparative fit index (CFI) = 0.95, Tucker-Lewis index (TLI) = 0.94, standardized root-mean-square residual (SRMR) = 0.06) displayed an excellent fit to the data. We further examined several alternative measurement models and compared them with the four-factor model. As shown in Table 1, the five-factor model fit our data better than other models, suggesting that our respondents could clearly distinguish the focal constructs.

### 4.3. Correlation Analysis

Table 2 shows the means, standard deviations, and correlations of the study variables. The results showed that job insecurity has a significant negative correlation with safety compliance (*r* = −0.18, *p* < 0.01), and Hypothesis 1 was supported. There is a significant negative correlation between job insecurity and safety participation (*r* = −0.26, *p* < 0.01), and Hypothesis 2 was supported.

Meanwhile, as shown in Table 2, job insecurity had a significant positive correlation with insomnia (*r* = 0.44, *p* < 0.01) and a significant negative correlation with work engagement (*r* = −0.36, *p* < 0.01). Insomnia had a significant negative correlation with work engagement (*r* = −0.27, *p* < 0.01) and a significant negative correlation with safety compliance (*r* = −0.23, *p* < 0.01) and safety participation (*r* = −0.25, *p* < 0.01). Work engagement had a significant positive correlation with safety compliance (*r* = 0.34, *p* < 0.01) and safety participation (*r* = 0.49, *p* < 0.01). These findings provide preliminary support for our hypothesis.

### 4.4. The Mediating Role of Insomnia and Work Engagement

First, we conducted a regression analysis of job insecurity with regard to safety compliance and safety behaviour, and the results show that job insecurity had a significant impact on safety compliance (*β* = −0.11, *p* < 0.001) and safety participation (*β* = −0.22, *p* < 0.001). Then, PROCESS 3.2 (International Business Machines Corporation, Armonk, NY, USA) was used to verify the multiple mediating effects of insomnia and work engagement. Bootstrapping was performed 5000 times with a 95% confidence interval. The results are shown in Table 3 and Figure 1.

Then, we conducted a mediation effect test on the model. As shown in Table 4 and Table 5, the 95% confidence interval of the indirect effect of job insecurity on safety compliance via insomnia was (−0.06, −0.01), and the 95% confidence interval of the indirect effect of job insecurity on safety participation via insomnia was (−0.07, −0.00). Hypotheses 3 and 4 were supported. The 95% confidence interval of the indirect effect of job insecurity on safety compliance via work engagement was (−0.08, −0.03), and the 95% confidence interval of the indirect effect of job insecurity on safety participation via work engagement was (−0.15, −0.07). Hypotheses 5 and 6 were supported. The 95% confidence interval of the indirect effect of job insecurity on safety compliance via insomnia and work engagement was (−0.02, −0.01), and the 95% confidence interval of the indirect effect of job insecurity on safety compliance via insomnia and work engagement was (−0.04, −0.01). Hypotheses 7 and 8 were supported.

## 5. Discussion

### 5.1. Theoretical Contributions

First, this study verifies the relationship between job insecurity and safety behaviour. In recent years, workplace safety behaviour has been a concern of most enterprises and scholars, and how to improve employee safety behaviours has become the focus of the current research. Most of the existing studies start at the organizational level investigating aspects such as leadership type [60,61], organizational climate [62], and organizational culture [63]. Other studies focus on individual factors such as safety knowledge [64] and safety motivation [65]. A hot topic in the current economic environment is the relationship between job insecurity and safe behaviour and its influencing mechanism, which are worthy of further exploration. The empirical results of this study show that employees’ work insecurity negatively affects safety behaviour; that is, the higher the employee’s sense of job insecurity, the lower the employee’s safety behaviour level will be. This result is illustrated by the theory of resource conservation theory. Conservation of resources theory posits that individuals have the instinct to acquire, maintain, protect and cultivate valuable resources [25]. Work offers daily income and living security for employees. It is an extremely important resource for employees to protect and maintain by working hard [22]. When this important resource is threatened, managing this threat will accelerate the consumption of employees’ recovery resources [66], which will result in a lack of redundant resources to monitor their behaviour and perform safety responsibilities, thus reducing their safety behaviour. Probst and Brubaker (2001) verify the relationship between job insecurity and safe behaviour with research on food processing industry samples [8]. However, with a continuous decline in oil prices and reforms in the oil industry, we believe that oil employees are more likely to experience work insecurity. The characteristics of the working environment also urgently call for higher-level safety behaviour. Therefore, it is meaningful to verify the relationship between job insecurity and the safety behaviour of oil workers. This study uses the theory of resource conservation to explain this relationship. It also shows the influence of employees’ subjective feelings about safety behaviour from another perspective. This study found that the impact of work insecurity on safe participation (β = −0.22, *p* < 0.001) was greater than the impact on safety compliance (β = −0.11, *p* < 0.001), which is in line with our prediction. This result also proves the value of studying safety behaviour in terms of two dimensions. Based on conservation of resources theory, we believe that the decrease in psychological resources caused by work insecurity is the reason for the decrease in employee safety behaviour. Moreover, the amount of resources required for safe compliance and safe participation are different. Safety compliance is a rigid requirement for employees, even in the condition of low psychological resources, and employees may have to comply under the pressure of the organization or supervisor. Safety participation, which is to some extent an altruistic behaviour, requires an increasing number of employees to actively participate and consumes an increasing amount of psychological resources. This finding is consistent with the previous research. In other words, there is an essential difference between safety compliance and safety participation [30], which confirms the importance of the dimensional exploration of security behaviour.

Second, this research verified the mediation effect of insomnia between job insecurity and safety behaviour. Insomnia is currently a common phenomenon despite the few studies that have been conducted on its relationship with work behaviour [36,67]. The results of this study show that employees’ job insecurity will lead to insomnia, thereby reducing their safety behaviours. In addition, the two dimensions of safe behaviour are studied separately, and this study found that insomnia has a significant mediation effect between job insecurity and safe participation or safe compliance. Job insecurity is a huge source of stress for employees, and this pressure will continue to exist after work, affecting employees’ sleep quality [16]. Poor sleep not only affects workers individual health problems, such as impairment of cognitive function, and current and subsequent affective diseases but also is associated with safe behaviour in terms of poor work performance, absenteeism, and increases in accidents at work [68]. Based on the theory of self-regulatory resources, job insecurity, as a source of stress, makes employees unable to fall asleep, and insomnia hinders employees’ recovery of their self-regulatory resources and causes them to have a lack of resources to regulate their behaviour [36], leading to the reduction of employees’ safety behaviours. Specifically, our results show that both safety compliance and safety participation must consume employees’ self-regulation resources; that is, after the insomnia caused by job insecurity, the reduction of employees’ self-regulation resources will not only cause them to ignore the organization’s safety rules and regulations, but it will also prevent them from actively carrying out certain behaviours that are beneficial to the safety of others and the organization. This result reveals the mechanism of safety behaviour and insomnia. This result not only reveals the mechanism of employees’ safe behaviour due to work insecurity but also verifies that employees’ insomnia can affect their behaviour in the workplace, which provides a new avenue for future research, that is, more attention must be paid to employees’ sleep quality.

Third, this research discovers the serial mediating role of insomnia and work engagement between job insecurity and safety behaviour. Employees’ sense of job insecurity increases employees’ insomnia problems and then reduces employees’ work engagement and ultimately leads to a reduction in employee safety behaviour. Kao’s research (2016) found that there is a negative correlation between insomnia and safe behaviours. Based on this, we further investigated and found a conduction effect of work engagement between these two factors [36]. Insomnia reduces employees’ work engagement and then reduces their safety behaviours. From the beginning, employees who perceive work insecurity think that it is hard to keep their job; thus, they are reluctant to subjectively devote too many resources to their current work. Insomnia causes employees to lack of the necessary resources to objectively devote themselves to work [36], and their work engagement decreases. The results of this study show the mediation effects of insomnia and work engagement on two dimensions and explain the importance of work engagement for safe behaviours. The safety compliance and safety participation of individuals with low work engagement will be reduced, leading to a great increase in potential safety hazards in the workplace. This discovery also allows this study to further clarify the mechanism of work insecurity on safe behaviours. Work engagement can be an important transmission mechanism that transmits employees’ pressure or negative feelings to their own safety behaviours.

### 5.2. Practical Significance

First, the empirical results of this research show that employees’ job insecurity will affect their own safety behaviours. Therefore, companies can increase employees’ safety behaviours in the workplace by reducing their work insecurity. It is more effective to deal with employees’ subjective perception compared with certain organizational factors (such as organizational climate and leadership types) [69]. For example, companies should establish channels for employees to obtain appropriate assistance such as an employee assistance program (EAP), consultation hotline or consultation seminar. Such channels can help solve any psychological or behavioural problems that might be encountered by employees or their family members, thereby reducing stress and work insecurity [5]. At the same time, companies may also provide training programs or team building for employees to maintain optimism, hope, flexibility and self-efficiency to reduce employees’ sense of work insecurity [70,71].

Moreover, given that insomnia plays a mediating role between employees’ work insecurity and safe behaviours, employees’ insomnia problems must be taken into serious consideration. Leaders must be considerate of employees and pay attention to their spiritual needs, such as sleep [72] and implement a “sleep-friendly” policy within the organization and create a “sleep culture” [73]. First, it is necessary to implement a flexible human resource management plan to avoid sleep deprivation caused by conflicts between employees’ work and life [72]. Second, it is necessary to strengthen sleep health education. Many employees have unhealthy lifestyles and sleeping habits. It is beneficial for people to establish healthy sleep habits to alleviate and eliminate sleep problems. Third, for employees with severe insomnia problems, leaders must engage in corresponding interference such as using emotional management, psychological guidance or anti-stress methods. Leaders are responsible for providing medical help when necessary to assist employees in restoring sleep health.

This study shows that work engagement is a very important part of the process of employees’ work insecurity in terms of safety behaviours. Increasing work engagement can help increase employees’ safety behaviours. The previous studies have shown that the proper authorization of work contributes to employees’ initiative and commitment [74]. Therefore, companies and leaders should appropriately increase employees’ work autonomy such as giving them more comfortable working spaces, more flexible working hours, and more free working boundaries. In addition, leaders should frequently communicate with employees and provide them with the mindset of a bright future to stimulate their working enthusiasm and improve their safety behaviour.

## 6. Limitations and Future Directions

First, our data collection was carried out during a single time period, which may be susceptible to common method variance [58]. Thus, future research can use multiple time points to collect data or adopt longitudinal designs to determine the causal direction of the relationships and effectively minimize the impact of common method variance [58].

Second, all our data were collected in a state-owned enterprise, which may to some extent affect the external validity of our study. The enterprise we chose is an oil company, which is a relevant case study regarding safe production. However, we still recommend that future research expand the sample size and explore whether the relationships identified here can be applied in other industries and in a cross-cultural context.

This study verified the relationship of job insecurity and safety behaviour and further explored the mediating role of insomnia and work engagement. However, we believe that there may be some boundary conditions in this mechanism. Thus, future research can focus on key organizational factors and explore whether they play a moderating role in this relationship.

## 7. Conclusions

This study revealed the relationship between job insecurity and safe behaviours, as well as the mediating effect of insomnia and job engagement. From the perspective of resource conservation theory, the study found that in the sample of employees of oil enterprises, employees who experienced more job insecurity were likely to have fewer safety behaviours. Job insecurity reduced employees’ safety behaviours by increasing their insomnia and reducing their work engagement.

## Figures and Tables

**Figure 1 ijerph-18-00581-f001:**
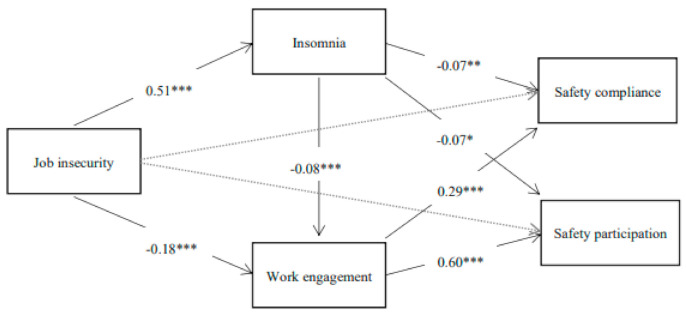
Multiple mediation effect model. (Note: *** *p* < 0.001; ** *p* < 0.01; * *p* < 0.05).

**Table 1 ijerph-18-00581-t001:** Results of confirmatory factor analysis of the measurement models.

Measurement Models	*χ* ^2^	*df*	*χ*^2^/*df*	RMSEA	CFI	TLI	SRMR
Five-factor (A, B, C, D, E)	1893.84	659	2.87	0.06	0.95	0.94	0.06
Four-factor (A, B, C, D + E)	2386.62	663	3.60	0.07	0.93	0.92	0.09
Three-factor (A, B + C, D + E)	4147.36	666	6.23	0.10	0.85	0.83	0.11
Two-factor (A + B + C, D + E)	4990.38	668	7.47	0.11	0.81	0.79	0.12
One-factor (A + B + C + D + E)	8062.86	669	12.05	0.14	0.68	0.64	0.15

Note: A = job insecurity, B = insomnia, C = work engagement, D = safety compliance, E = safety participation.

**Table 2 ijerph-18-00581-t002:** Scale descriptive statistics and correlations among study variables (N = 568).

	M	SD	1	2	3	4	5	6	7	8	9	10
1 gender	1.43	0.50	-									
2 age	4.23	1.13	−0.15 **	-								
3 marriage	1.12	0.41	0.02	−0.18 **	-							
4 education	2.90	0.93	0.21 **	−0.34 **	0.07	-						
5 years of working	4.76	0.64	−0.14 **	0.59 **	−0.19 **	−0.28 **	-					
6 job insecurity	3.47	1.22	−0.05	0.10 *	0.03	−0.31 **	0.13 **	-				
7 insomnia	2.71	1.44	−0.18 **	0.07	0.06	−0.22 **	0.10 *	0.44 **	-			
8 work engagement	3.67	0.76	0.11 *	0.05	−0.09 *	0.06	−0.02	−0.36 **	−0.29 **	-		
9 safety compliance	6.45	0.76	0.11 **	0.06	−0.04	−0.05	0.01	−0.18 **	−0.23 **	0.34 **	-	
10 safety participation	6.06	1.02	0.03	−0.03	−0.01	0.02	−0.03	−0.26 **	−0.25 **	0.49 **	0.68 **	-

Notes: N = 568. Gender was coded “1” for women and “2” for men. Age was coded “1” for under 25 years old, “2” for 26–30 years old, “3” for 31–35 years old, “4” for 36–40 years old, “5” for 41–50 years old, and “6” for over 50 years old. Marriage was coded “1” for married and “2” for unmarried. Education was coded “1” for junior high school diploma, “2” for high school diploma, “3” for associate degree, “4” for undergraduate diploma, and “5” for master diploma. Years of work were coded “1” for less than 1 year, “2” for 1–3 years, “3” for 4–6 years, “4” for 7–9 years, and “5” for more than 10 years. * *p* < 0.05; ** *p* < 0.01.

**Table 3 ijerph-18-00581-t003:** The mediating role of insomnia and work engagement.

	Outcome: Criterion: Insomnia			Outcome: Work Engagement			Outcome: Safety Compliance			Outcome: Safety Participation		
	*β*	SE	*t*	*β*	SE	*t*	*β*	SE	*t*	*β*	SE	*t*
Gender	−0.46	0.11	−4.30 ***	0.10	0.06	1.67	0.09	0.06	1.44	−0.09	0.08	−1.13
Job insecurity	0.51	0.04	11.71 ***	−0.18	0.03	−6.63 ***	−0.01	0.03	−0.45	−0.05	0.04	−1.43
Insomnia				−0.08	0.02	−3.52 ***	−0.07	0.02	−2.81 **	−0.07	0.03	−2.34 *
Work engagement							0.29	0.04	6.85 ***	0.60	0.05	11.39 ***
R^2^	0.22			0.15			0.14			0.26		
F	80.28 ***			34.40 ***			22.66 ***			49.59 ***		

Note: *** *p* < 0.001; ** *p* < 0.01; * *p* < 0.05.

**Table 4 ijerph-18-00581-t004:** Indirect effects of job insecurity on safety compliance based on 5000 bias-corrected bootstrapped samples.

	Effect	BootSE	BootLLCI	BootULCI	Mediation Proportion
Total indirect effect: JI→SC	−0.10	0.02	−0.13	−0.07	88.58%
Indirect effect via insomnia	−0.03	0.01	−0.06	−0.01	30.76%
Indirect effect via WE	−0.05	0.01	−0.08	−0.03	46.85%
Indirect effect via insomnia and WE	−0.01	0.00	−0.02	−0.01	10.97%

Note: JI = job insecurity, SC = safety compliance, WE = work engagement.

**Table 5 ijerph-18-00581-t005:** Indirect effects of job insecurity on safety participation based on 5000 bias-corrected bootstrapped samples.

	Effect	BootSE	BootLLCI	BootULCI	Mediation Proportion
Total indirect effect: JI→SP	−0.17	0.02	−0.22	−0.12	77.03%
Indirect effect via insomnia	−0.04	0.02	−0.07	−0.00	16.19%
Indirect effect via WE	−0.11	0.02	−0.15	−0.07	49.26%
Indirect effect via insomnia and WE	−0.03	0.01	−0.04	−0.01	11.57%

Note: JI = job insecurity, SP = safety participation, WE = work engagement.

## Data Availability

The datasets during and/or analyzed during the current study are available from the corresponding author on reasonable request.

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
