# Peer review of "Job Insecurity and Safety Behaviour: The Mediating Role of Insomnia and Work Engagement"

_ijerph, 2021, doi:10.3390/ijerph18020581_

Round 1

Reviewer 1 Report

Although there is broad interest in mediation analysis from many areas of psychology and other fields, I found some parts of the present study design and manuscript writing problematic.

Specific comments:

  1. "Given that the economic and technical development that has taken place over the last few decades has led to improvements in employees’ workplace safety (Guo & Yiu, 2016; Hinze, Hallowell, & Baud, 2013), job insecurity is still challenging and poses a critical problem in several high-risk industries such as petrochemical enterprises" - this is an extremely convoluted sentence. Please rephrase.
  2. "Many studies have argued that unsafe behaviour is ascribed to feelings of unsafety" - do you mean feelings of 'insecurity'?
  3. "despair and indolence" - 'indolence' is an inappropriate word choice here. 
  4. "568 employees from an oil enterprise located in China" - did they all work in the same location? How did the authors decide on recruitment?
  5. How was the questionnaire developed? Was it with reference to international studies on the subject? Did the authors conduct any pilot test for the questionnaire?
  6. The authors need to provide more detail on how the questionnare was disseminated and what platform was the survey conducted over? Also, if the surveys were conducted online, did the authors take steps to guard against duplicate responses to your survey? E.g. IP filtering?
  7. On average, how long does it take for a respondent to complete the questionnaire?
  8. The response rate seems abnormally high. What were some of the reasons for this? Were the participants incentivized to participate in the study? How many reminders were sent to the participants? When studies experience low response rates, they will tend to be biased toward the positive. Conversely, studies with high response rates may be negatively biased.
  9. When presenting results, please provide the exact n and accompanying percentage, e.g. "83.1% (n=?) had worked for more than 10 years."
  10. Because confidence limits are important for understanding effects, confidence limits based on the distribution of the product or the bootstrap are recommended.
  11. The underlying data should be made publicly available. If this was not possible, please provide a reason why.
  12. Copies of the full questionnaire should be appended.

Author Response

Thank you for taking time to provide this in-depth review of our paper. This valuable information has allowed us to improve the manuscript. Below, we clarify the changes that have been made.

Reviewer 2 Report

This paper explores the relationships between job insecurity and safe behaviors as well as the effects of insomnia and level of job engagement on job security and safe behaviors.  This study, completed in an oil industry setting, is particularly important for guiding employers to assure safety for employees including providing supports for addressing personal needs (job security) and high participation in safe procedures in the work setting. As mentioned by the authors, exploring these same principles in other work settings would strengthen the findings and guide employers in support of their employees to learn and follow safe practices. The practical significance of these findings are the important recommendations for leaders in various work settings. 

I suggest several edits to the language:

line 87 change "be difficult" to "have difficulty"

line 105 edit to 2 separate sentences: . after "safety policy."  New sentence beginning with "Safety participation..."

line 111 change "than" to "regarding"

line 395 insert "of" after "lack"

line 416 insert "of" after "lack"

Author Response

Response to Reviewer

Thank you for taking time to provide this in-depth review of our paper. This valuable information has allowed us to improve the manuscript. Below, we clarify the changes that have been made.

This paper explores the relationships between job insecurity and safe behaviors as well as the effects of insomnia and level of job engagement on job security and safe behaviors.  This study, completed in an oil industry setting, is particularly important for guiding employers to assure safety for employees including providing supports for addressing personal needs (job security) and high participation in safe procedures in the work setting. As mentioned by the authors, exploring these same principles in other work settings would strengthen the findings and guide employers in support of their employees to learn and follow safe practices. The practical significance of these findings are the important recommendations for leaders in various work settings. 

I suggest several edits to the language:

line 87 change "be difficult" to "have difficulty"

line 105 edit to 2 separate sentences: . after "safety policy."  New sentence beginning with "Safety participation..."

line 111 change "than" to "regarding"

line 395 insert "of" after "lack"

line 416 insert "of" after "lack"

Response: Thank you for your kind advice. In the revised manuscript, we have edited the language and marked this in red. The details are as follows:

line 87 In the context of lack of psychological resources, employees will have difficulty time coping with the complex and changeable environment and dangerous events in the workplace, and reduce the compliance with safety regulation[27]. [line 82]

line 105 Safety compliance means that employees can carry out production operations according to the enterprise’s safety policy. Safety participation means that they can spontaneously engage in behaviours that are conducive to corporate safety at work, including behaviours such as assisting with colleagues, attending safety conferences, and taking the initiative to improve safety in the workplace. [line 97]

line 111 Based on the conservation of resources theory, we believe that safety participation requires employees to be more proactive regarding safety compliance (Li, Jiang, Yao, & Li, 2013). [line 103]

line 395 Based on the theory of self-regulatory resources, insomnia will hinder employees’ recovery of their self-regulatory resources and cause them to have a lack of resources to regulate their behaviour [36]. [line 399]

line 416 Insomnia causes employees to lack of the necessary resources to objectively devote themselves to work [36], and their work engagement decreases. [line 418]

Reviewer 3 Report

1. Introduction: the authors should consider deleting or rephrasing the sentence "main reason for these accidents is the improper behaviors of employees." I think given the nature of occupational health research and NIOSH goal towards TWH, it is unjust and incorrect to assume that safety accidents on the job are solely the fault of employees. Safety accidents can also be do to certain organizational barriers and practices that keep employees from being able to work in a safe environment. The authors statement is also based on a single study.

2. Introduction: the authors should consider talking about organizational barriers and/or practices that prevent employees from working in a safe environment in their introduction section.

3. Theory and hypotheses: authors should consider rewording the following "In the context of lack of psychological resources, employees will be difficult to cope with.." to "In the context of lack of psychological resources, employees will have a difficult time coping with.."

4. Theory and hypotheses: authors should consider rewording the following "resource conservation theory" to "conservation of resources theory."

5. The mediating effect of insomnia: authors should talk a little bit more about how poor sleep is affected by ones work demands and environment. Additionally, the authors should consider rewording the following "This fear and worry will cause employees to experience tremendous psychological stress after work (Burgard & Ailshire, 2009), leading to insomnia at night" to "This fear and worry will cause employees to experience tremendous psychological stress after work (Burgard & Ailshire, 2009), which can lead to insomnia at nigh."

6. It would be helpful if the authors had included a figure with their hypothesized relationships as they walk the readers through their theoretical models.

7. Methods: Sample and Procedures: authors should consider changing the word "secret" to "confidential."

8. Methods: Authors did a good job explaining their methods. However, what is missing is their statistical analysis approach. What statistical technique did you use to determine the associations between your study hypotheses? Was is SEM or did you use Ordinary Least Squares Regression?

9. Discussion: authors should consider changing "theory of resource conservation theory" to "conservation of resources theory." If the authors are using Hobfolls theory (1989) it is referred to as conservation of resources theory only.

10. Discussion: Because this is a cross-sectional study, authors should  refrain from using causal language such as " insomnia will affect their behaviour in the workplace,which provides a new avenue for future research, that is, more attention must be paid to employees’ sleep quality." The authors should instead say that their results showed that insomnia can affect..or that insomnia is associated with...

11. Discussion: The authors should frame their research findings on the topic of insomnia within the context of similar research done on this topic.

12. Conclusion: the conclusion of this paper needs work. The authors repeat their study findings in the conclusion section. The conclusion section should be a short summary of the overall paper.

Author Response

(The authors gave the same response as above.)

Round 2

Reviewer 1 Report

There are still rampant grammatical and language issues throughout the manuscript.

Specific comments:

  1. Please change "However, Job insecurity is still challenging" to "However, job insecurity is still an issue".
  2. Please change "organizational environmental factors" to "organizational and environmental factors".
  3. The style of citation is inconsistent, e.g." places employees in a stressful situation[12] (Giunchi, Emanuel, Chambel, & Ghislieri, 2016)." Please standardize throughout according to the journal guidelines.
  4. Much of the discussion is centered around the conservation of resources theory. With regard to burnout and general stress, several other theories and research have explored how the loss of resources impacts one’s level of perceived stress. These should be outlined as well.

Author Response

Dear Professor,

 Thank you for your kind work. Please see the attachment.
